# Vericiguat in Heart Failure: Characteristics, Scientific Evidence and Potential Clinical Applications

**DOI:** 10.3390/biomedicines10102471

**Published:** 2022-10-03

**Authors:** Francesca Vannuccini, Alessandro Campora, Maria Barilli, Alberto Palazzuoli

**Affiliations:** 1Cardiovascular Diseases Unit, Cardio-Thoracic and Vascular Department, “Le Scotte” Hospital Siena, University of Siena, 53100 Siena, Italy; 2Department of Medical Biotechnologies, Division of Cardiology, “Le Scotte” Hospital Siena, University of Siena, 53100 Siena, Italy

**Keywords:** heart failure, treatment, vericiguat, soluble guanylate cyclase

## Abstract

Despite recent advances in heart failure (HF) management, the risk of death and hospitalizations remains high in the long term. HF is characterized by endothelial dysfunction, inflammation and increased oxidative stress, due to a reduction in the activity of the nitric oxide (NO)-soluble guanylate cyclase (sGC)-cyclic guanosine monophosphate (cGMP) signaling pathway. All these factors contribute to direct damage at the myocardial, vascular and renal level. Vericiguat restores the deficiency in this signaling pathway, through stimulation and activation of sGC, aiming to increase cGMP levels, with a reduction in HF-related oxidative stress and endothelial dysfunction. Two main clinical trials were developed in this setting: the SOCRATES-REDUCED phase II study and the VICTORIA phase III study. They found that vericiguat is safe, well tolerated and effective with an absolute event-rate reduction in patients affected by HF with reduced ejection fraction (HFrEF) and recent cardiac decompensation. In patients with HF with preserved ejection fraction (HfpEF), the SOCRATES-PRESERVED trial demonstrated an improvement in quality of life and health status, but the proven beneficial effects with vericiguat are still limited. Further studies are needed to correctly define the role of this drug in heart failure syndromes. Our paper reviews the potential applications and pharmacological characteristics of vericiguat in HFrEF and HFpEF.

## 1. Introduction

Heart failure (HF) is a major global heart pathology, affecting an average of 64.3 million people worldwide [1]. Moreover, its prevalence is increasing due to aging of the general population and better outcomes after acute cardiovascular events. Although new therapies and management strategies have reduced mortality and morbidity, the prognosis for these patients remains poor. It is estimated that only 50% of patients survive after 5 years from their initial diagnosis. Moreover, repeated hospitalizations and the need for supplemental parenteral therapy during frequent exacerbations indicate an impaired quality of life and worse prognosis [2].

The increase in hospitalization rate and mortality associated to the initial diagnosis of HF justifies the research of new therapeutic agents. Many treatment options are now available for the management of HF with reduced ejection fraction (HFrEF), based on large randomized controlled trials and accessible in the American Heart Association/ American College of Cardiology and the European Society of Cardiology guidelines [2,3,4]. The milestones of drug treatment for HfrEF are angiotensin-converting enzyme inhibitors (ACEi), angiotensin receptor blockers (ARBs), beta-blockers and mineralocorticoid receptor antagonists (MRAs) with a Class I recommendation. Furthermore, sacubitril/valsartan, an angiotensin receptor neprilysin inhibitor (ARNI), and the sodium-glucose cotransporter 2 inhibitors (SGLT2i) empagliflozin and dapagliflozin have been added to the list of disease-modifying therapies [5].

Vericiguat is currently considered a second step of treatment in patients who remained symptomatic despite optimized medical therapy (OMT), to improve outcomes in HFrEF (class of recommendations IIb) [3].

In this review, we analyze the characteristics of vericiguat, some of the new evidence that has emerged in the latest studies and the next steps to take in order to consolidate the clinical use of this new weapon, with a focus on the clinical trials literature.

## 2. Mechanism of Drug Action

Vericiguat is a drug that stimulates the cyclic guanosine monophosphate (cGMP) pathway through direct and indirect stimulation of soluble guanylate cyclase (sGC) [6]. The downstream effects of this stimulation pathway are smooth muscle cell relaxation, reduction in hypertrophy, inflammation and fibrosis [7].

The nitric oxide (NO)-sGC-cGMP pathway begins with NO production by vascular endothelial cells. NO is synthesized from L-arginine by three nitric oxide synthases, among which endothelial nitric oxide synthase (eNOS) plays a major role. NO diffuses rapidly into vessel smooth muscle cells, binds to the heme subunit of sGC and catalyzes the conversion of guanosine triphosphate (GTP) into the second intracellular messenger, cGMP [8]. CGMP interacts with three types of intracellular proteins: cGMP-dependent protein kinases, cGMP-regulated ion channels and phosphodiesterases (PDEs) [9]. Subsequently, these transduction cascades mediate various physiological and tissue-protective effects, including smooth muscle relaxation, inhibition of smooth muscle proliferation, leukocyte recruitment and platelet function [7,10].

In HFrEF, tissue hypoperfusion caused by a reduction in cardiac output induces inflammation and oxidative stress, leading to a decrease in NO bioavailability and decreased activity of cGMP [5]. Reduced sGC activity is associated with coronary microvascular dysfunction, cardiomyocyte stiffness and interstitial fibrosis, fundamental elements that lead to the progression of myocardial dysfunction [7]. Therefore, sGC stimulators, such as vericiguat, may be particularly effective in this condition, counteracting endothelial dysfunction and increased oxidative stress through cGMP elevation by a double pathway for enzyme activation (Figure 1) [7].

CGMP is also enhanced by others signaling pathways. The natriuretic peptides (NPs, atrial natriuretic peptide and the B-type natriuretic peptide) increase cGMP through activation of membrane-bound guanylate cyclase (particularly guanylate cyclase, pGC) [11,12]. Some therapeutic strategies, which act in the NP-pGC-cGMP pathway, have been evaluated, such as synthetic NPs and NP analogs (nesiritide, ularitide) and ARNI (sacubitril/valsartan), that increase NPs through inhibition of neprilysin. In clinical trials, the use of sacubritil/valsartan has been shown to significantly improve outcomes in HF [13,14].

Degradation of cGMP in GMP is catalyzed by seven, differentially expressed PDE families [9]. PDE inhibitors, such as milrinone and enoximone (PDE-3 inhibitors) and sildenafil (PDE-5 inhibitors), have also been evaluated as therapeutic strategies in the context of HF [12]. PDE-5 inhibitors improve contractile function in systolic HF and reduce remodeling of the left ventricle [5]. However, to date, no randomized clinical trial has demonstrated an improved outcome in HF with the use of PDE inhibitors [7,12].

**Figure 1 biomedicines-10-02471-f001:**
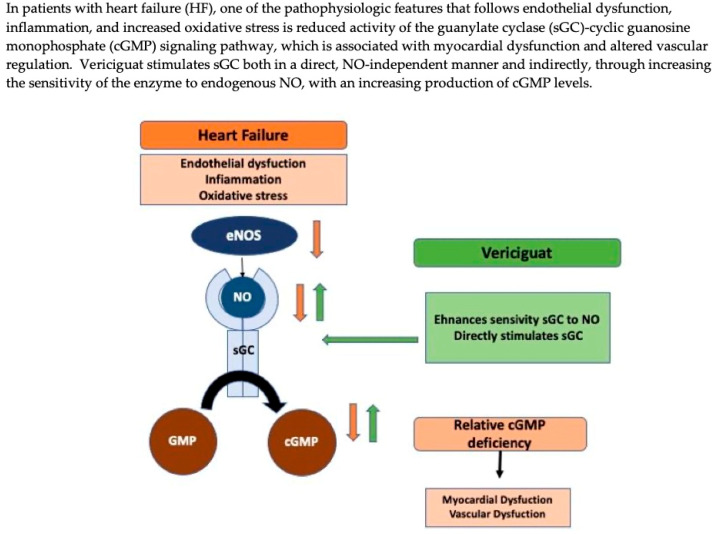
Mechanisms of action of vericiguat. cGMP= cyclic guanosine monophosphate; eNOS = endothelial nitric oxide synthase; GMP = guanosine monophosphate; NO = nitric oxide; sGC = soluble guanylate cyclase. Adapted with permission from Ref. [12]. Copyright 2021, Arrigo Mattia.

## 3. Pharmacological Proprieties of Vericiguat

Vericiguat is a weakly basic drug with a low water solubility and high intestinal permeability (class II according to the Biopharmaceutics Classification System) [15].

In six phase I studies, analysis on healthy volunteers [16], carried out with the aim of evaluating the safety, tolerability, pharmacodynamic and pharmacokinetic of vericiguat demonstrated that its chemical structure exhibits an excellent oral bioavailability (93%) with a long half-life (18–22 h) that allows oral administration once daily. It also has a high pharmacokinetic stability and a lower variability after administration with food [5,16]. Vericiguat is a low-clearance drug (1.6 L/h in healthy volunteers and 1.3 L/h in patients with HFrEF) with a plasma protein binding of approximately 98%, serum albumin being the main binding component without alterations in the case of renal or hepatic impairment [17].

In healthy subjects, after an oral administration of vericiguat, about 53% of the dose is excreted in the urine and 45% is excreted in the feces. For this reason, as demonstrated in phase I clinical trials during the titration regimen (from 0.5 mg to 15 mg daily), the drug can be administered without dose adjustments in patients with renal impairment up to an estimated glomerular filtration rate (eGFR) of 15 mL/min/1.73 m^2^ or with moderate liver disease [16].

Vericiguat is mainly metabolized by glucuronidation through uridine diphosphate-glucuronosyltransferase (UGT 1A9 and UGT 1A1) to N-glucuronide M-1, which is pharmacologically inactive against sGC. A small part of the drug (<5%) is metabolized by the CYP clearance pathway [15,17]. In vitro studies in phase I drug–drug interaction studies revealed that vericiguat shows a low potential for pharmacological interactions: the main molecule and its N-glucuronide metabolite do not act as inhibitors of major CYP isoforms, UGT isoforms, major transport proteins or as inductors of cytochrome P450 [15,17].

In phase I clinical trials, the most frequent drug-related adverse effects were nervous system disorders (headache, postural dizziness, hypotension) or gastrointestinal system disorders (dyspepsia, nausea). These adverse effects are explained by vericiguat’s action of smooth muscle relaxation, both in the gastrointestinal tract and vessels. In these studies, no clinically relevant alterations in laboratory parameters were observed [16]. In the randomized, placebo-controlled phase II clinical trial SOCRATES (SOluble guanylate Cyclase stimulatoR in heArT failurE Study)-REDUCED [18], this drug was found to be safe, well tolerated and without significantly worsening hemodynamic parameters compared with placebo. The most frequent adverse effects reported were symptomatic hypotension, syncope, anemia, dyspepsia, nausea and headache. The mechanism underneath the development of anemia is not yet completely understood but has been reported also with the use of another sGC stimulator (riociguat). The incidence of serious adverse events (treatment-emergent acute kidney injury, syncope or hypotension) was slightly higher in the placebo group (32.6%) compared with the treated group (range 22% to 28.9%), with the lowest incidence reported for subjects in the lower-dose groups [18]. The same most frequent adverse effects, with an incidence greater than 2%, occurred in the phase III VICTORIA [19] (Vericiguat Global Study in Subjects with Heart Failure with Reduced Ejection Fraction) clinical trial. However, symptomatic hypotension (9.1% vs. 7.8%) and syncope (4% vs. 3.5%) were more common in the patients receiving vericiguat than in those receiving placebo, while general adverse events were slightly higher in the placebo group (80.5% vs. 81.0%) [20]. Systolic blood pressure declined slightly in both treatment arms (about 5 mmHg in systolic blood pressure, compared with baseline values) during the first 16 weeks of the study and then returned to baseline levels [19]. Anemia developed more frequently in patients in the vericiguat group than in the placebo group (in 7.6% and 5.7%), but only in 1.6% of patients who presented anemia with vericiguat was it considered a severe adverse event [19].

## 4. Main HF Clinical Trials

### 4.1. Vericiguat in HFrEF

Two main clinical trials provided the initial evidence for the use of vericiguat in the context of HFrEF: the phase II study SOCRATES–REDUCED [18] and the phase III study VICTORIA [19].

The SOCRATES-REDUCED trial was designed to evaluate the tolerability and the optimal dose of vericiguat in patients with chronic HF and reduced left ventricular ejection fraction (LVEF), in addiction to standard therapy. Thus, 456 patients with chronic HFrEF (LVEF < 45%) were enrolled (NYHA functional classes II–IV) with an episode of worsening HF within 4 weeks of randomization, defined by symptoms or signs of congestion that required hospitalization or outpatient administration of intravenous (IV) diuretics together with an elevated level of B-natriuretic peptide (BNP) ≥ 300 pg/mL or N-terminal pro-B natriuretic peptide (NT-proBNP ≥ 1000 pg/mL if in sinus rhythm; BNP ≥ 500 pg/mL or NT-proBNP ≥ 1600 pg/mL if in atrial fibrillation) (Table 1) [18]. The mean LVEF of the patients enrolled was 29.6% and all of them were already on treatment with HF guideline-directed medical therapy for at least 1 month before HF hospitalization (HFH) or outpatient IV diuretics administration. None of the patients received ARNI or SGLT2i therapies. They were randomized to placebo or to vericiguat on one of the four target doses (1.25 mg, 2.5 mg, 5 mg or 10 mg once daily). Planned total treatment duration was 12 weeks, followed by a safety follow-up at 16 weeks after randomization [18,21].

The study found that over a 12-week period, the change in log-transformed NT-proBNP levels (primary endpoint) was not considerably different in the pooled vericiguat group compared with the placebo group (*p* = 0.15). However, a secondary exploratory analysis of the primary endpoint showed that higher doses of vericiguat were associated with a greater reduction in NT-proBNP values (*p* < 0.02). Additional echocardiographic analyses showed that patients in the vericiguat 10 mg group had an increase in LVEF at 12 weeks compared with placebo (+3.7% vs. +1.5%; *p* = 0.02), but no significant differences were found in the change in left ventricular end-diastolic volume (LVEDV) or left ventricular end-systolic volume (LVESV) in the two groups. HFH were accounted among the secondary endpoints along with blood pressure, heart rate and changes in the levels of multiple biomarkers [18]. Despite no differences encountered for the secondary endpoints, patients receiving the two highest doses of vericiguat experienced a reduced rate of HFH (9.9% vericiguat vs. 17.4% placebo), suggesting a dose–response relationship with higher doses of vericiguat. No significant changes in mean blood pressure (systolic and diastolic) and mean heart rate from baseline to 12 weeks were found between the clusters (all *p* ≥ 0.57) (Table 2) [18].

The recent phase III randomized, multicenter VICTORIA trial, where high-risk HF patients were treated with vericiguat, showed a considerable reduction in the composite primary endpoint of death from cardiovascular causes or first HFH [8]. The secondary outcomes were the components of the primary outcome, first and subsequent HFH, a composite of death from any cause or first HFH and death from any cause [22].

The VICTORIA trial enrolled 5050 subjects (≥18 aa) with chronic HFrEF (NYHA functional classes II–IV) with reduced LVEF (defined as LVEF < 45% within 12 months before randomization) and a recent episode of worsening HF defined by symptoms of HF or signs of congestion that required hospitalization or outpatient administration of IV diuretics and an elevated natriuretic peptide level within 30 days before randomization (BNP ≥ 300 pg/mL or NT-proBNP ≥ 1000 pg/mL if in sinus rhythm; BNP ≥ 500 pg/mL or NT-proBNP ≥ 1600 pg/mL if in atrial fibrillation) (Table 1). They were categorized into three groups based on the timing of deterioration: those hospitalized within 3 months before randomization, those hospitalized 3 to 6 months before randomization and those receiving outpatient IV diuretic therapy within 3 months before randomization. The estimated eGFR of the patients enrolled was up to 15 mL/min/1.73 m^2^ within 30 days before randomization, the mean LVEF was 29% and all of them were already on treatment with HF guideline-directed OMT [22,23].

Patients were randomized to placebo or vericiguat 2.5 mg once daily, up-titrated to 5 mg and then to 10 mg at 2-week intervals. This titration criterion is based on evaluation of mean systolic blood pressure and clinical symptoms at 2-week intervals. In 89.2% of cases, the target dose of vericiguat was reached [19].

The study showed that in patients treated with vericiguat, the incidence of primary outcomes was lower than in patients treated with placebo. Specifically, the incidence of first HFH or death from cardiovascular causes occurred in 897 patients (35.5%) in the vericiguat group and in 972 patients (38.5%) in the placebo group (HR 0.90; CI 0.82–0.98; *p* = 0.02). There were 1223 total HFH, including first and recurrent events (38.3 events per 100 patient-years) in the vericiguat group compared with 1336 total HFH recorded (42.4 events per 100 patient-years) in the placebo group (*p* = 0.02). Death from any cause or first HFH (a composite secondary outcome) occurred in 957 patients (37.9%) in the vericiguat group and in 1032 patients (40.9%) in the placebo group (HR 0.90; CI 0.83–0.98; *p* = 0.02) (Figure 2, Table 2) [19,23].

During follow-up (median duration of 10.8 months), treatment with vericiguat was significantly associated with a 10% reduction in the primary outcome. This finding translates into an absolute event rate reduction of 4.2 events per 100 patient-years; in other words, it is possible to assert that it is necessary to treat 24 patients with vericiguat for 1 year to prevent a primary event. This important result was obtained in patients who were already undergoing guideline-based OMT. Ultimately, the incidence of death from any cause showed no difference between the two groups (Table 2) [19].

Some significant differences emerge comparing the characteristics of these patients with those of previous clinical trials, specifically regarding the basal risk profile and the severity of HF at randomization. Patients included in the VICTORIA trial were older, less stable and with worse clinical conditions compared with the PARADIGM-HF [14] and DAPA-HF [24] study populations. All patients included in the VICTORIA trial had a recent episode of HFH, while the PARADIGM-HF and DAPA-HF trials only had rates of HFH of 62.5% and 47.5%, respectively. Furthermore, patients included in the VICTORIA study exhibited higher values of NT-proBNP (2816 pg/mL) compared to the PARADIGM-HF and DAPA-HF trials (1608 and 1437 pg/mL respectively) and a greater number of patients (41%) were in NYHA class III- IV compared to the 25% and 32% of the patients in the PARADIGM-HF and DAPA-HF trials, respectively [14,24,25,26].

A comparative analysis of the absolute risk reduction between these studies suggested that the absolute risk reduction was similar between vericiguat and SGLT2i, whereas the risk reduction was greater for vericiguat compared to sacubitril/valsartan [25,27].

The results regarding the primary outcome were consistent in the different subgroups of patients analyzed (Table 3).

Subgroup analysis showed that patients randomized for a longer time since last HFH had a greater benefit. Therefore, the benefit of vericiguat was independent from the baseline patient’s treatment of HF, either when analyzed alone or in combination. Moreover, the combination of vericiguat with the concomitant use of sacubritil/valsartan did not demonstrate an additional benefit (use of sacubritil/valsartan: HR 0.88 vs. no use of sacubritil/valsartan: HR 0.90). Finally, the efficacy of vericiguat on LVEF value demonstrated a greater benefit in patients with a moderate reduction in ejection fraction (LVEF < 40%) (Table 3) [19].

Voors AA et al. [28] evaluated the relationship between vericiguat efficacy and changes in renal function, showing a beneficial effect of this drug, regardless of the patient’s renal function. This analysis verified that the beneficial effects of vericiguat on the primary endpoint of cardiovascular death or HFH were maintained across all the eGFR spectrum at baseline and that they were comparable in both patients who developed a reduction in renal function and those who maintained the initial values.

Analyses of the drug according to the NT-proBNP levels at randomization found that major benefits were more evident in patients with NT-proBNP levels < 8000 pg/mL. This was further amplified in patients with <4000 pg/mL levels [29].

All these findings may suggest some favorable effect on the outcome in some specific subgroups of patients with advanced HF and prone to recurrent HFH.

### 4.2. Vericiguat in HFpEF

Due to its pharmacodynamic features, vericiguat has been tested in HF with preserved ejection fraction (HFpEF). Two main clinical trials have been published in this field: the phase II study SOCRATES–PRESERVE [30] and the phase III study VITALITY (Evaluate the Efficacy and Safety of the Oral sGC Stimulator Vericiguat to Improve Physical Functioning in Daily Living Activities of Patients With Heart Failure and Preserved Ejection Fraction) [31].

The phase II SOCRATES-PRESERVED was a randomized, double-blind, placebo-controlled, dose-finding clinical trial in 477 patients with symptomatic chronic HF (NYHA functional classes II–IV) with preserved LVEF (defined as LVEF ≥ 45%) and a recent episode of worsening HF within 4 weeks, which required hospitalization or IV outpatient’s diuretic treatment, together with an elevation in natriuretic peptide levels at randomization (BNP ≥ 100 pg/mL or NT-proBNP levels ≥ 300 pg/mL if in sinus rhythm; BNP ≥ 200 pg/mL or NTproBNP ≥ 600 pg/mL if in atrial fibrillation) [26,30].

This study was designed to analyze safety, tolerability and pharmacological proprieties of four different doses of vericiguat for 12 weeks. The primary outcome of the SOCRATES-PRESERVED included the change in log-transformed NT-proBNP levels (one-sided *p* = 0.90, two-sided *p* = 0.20) and left atrial volume over 12 weeks of treatment (one-sided *p* = 0.81, two-sided *p* = 0.37). It was not significantly different in the pooled vericiguat treatment group compared to the placebo group. Despite the absence of effect on these markers, an improvement in quality of life and health status, as assessed by the Kansas City Cardiomyopathy Questionnaire (KCCQ), was observed in patients treated with the two highest doses of vericiguat (mean difference 19.8 points from baseline) compared to placebo (mean difference 9.2 points from baseline) (Table 4) [30,32].

The VITALITY study analyzed the efficacy and safety of vericiguat on quality of life and exercise tolerance in patients with HFpEF assessed by KCCQ scores and the six-minute walk test (6MWT), respectively. 

The study included 789 patients with chronic HFpEF (NYHA functional class II--III) and a preserved LVEF (defined as LVEF ≥ 45%) with an episode of decompensation in the previous 6 months that required hospitalization or outpatient administration of IV diuretics and elevated natriuretic peptides. In this randomized study, there were three arms, two in which vericiguat was tried up to 15 mg or 10 mg and one who received placebo. After 24 weeks of treatment, there were no significant differences between the two groups concerning KCCQ scores or the six-minute walk test (Table 4) [26,31].

The cGMP pathway plays a role in HFpEF pathophysiology. Improved characterization of cGMP signaling and its relation to cardiac function revealed multiple options for targeted therapy [33,34]. To date, no large phase III HFpEF trial has definitively tested the effects of a pharmacologically mediated increase in cGMP activity. This needs to be further investigated [33].

## 5. Future Perspectives

The 2021 European HF guidelines introduced, in addition to the recommended first-line therapy, including ACEi, beta-blockers and MRA, the use of ARNI and SGLT2i for patients with HFrEF, to reduce the risk of HFH and death. Vericiguat administration is suggested as a second-line treatment in patients who remain symptomatic despite OMT, mainly after a recent episode of decompensation [2].

These recommendations are mainly based on pivotal clinical trials. Nevertheless, in our opinion, a tailored approach, even beyond clinical practice guidelines, needs to be assessed based on specific clinical biochemical and instrumental features: (1) the trend of NT-proBNP value, since treatment with vericiguat has shown a greater benefit in patients with NT-proBNP < 8000 pg/mL, particularly if <4000 pg/mL; (2) the LVEF, with greater benefit in patients with mild reduction of ejection fraction (LVEF < 45%, particularly if LVEF < 40%); and (3) time since the last HFH, with greater benefit when longer time had passed since the last HFH. Moreover, vericiguat has been shown to be suitable in patients with chronic kidney disease and it can be used independently of the patient’s baseline treatment. Therefore, in patients with HFrEF with these characteristics, we could hypothesize a therapeutic strategy based on additional use of vericiguat as a first-line treatment, along with the other drugs currently indicated in the guidelines (ACEi or ARNI, beta-blockers, MRA and SGLT2i). However, further studies and clinical evidence are needed to demonstrate the efficacy of its use in this setting of patients.

To date, studies on vericiguat are limited by a short follow-up period and a small number of patients compared with other clinical trials in HF. In the PARADIGM-HF trial, 8442 patients were enrolled compared with 5050 patients enrolled in the VICTORIA trial and the median follow-up values were 27 months and 18 months in PARADIGM-HF and DAPA-HF trials, respectively, compared with 11 months in the VICTORIA trial [14,22,25,26]. We can, therefore, assume that a longer follow-up period might be useful to show additional data on absolute cardiovascular risk reduction beyond those already obtained.

Notably, compared with other pivotal studies conducted in chronic HF, the patients randomized in the VICTORIA trial appear less stable, with more severe conditions and with a higher risk of cardiovascular events, as evidenced by the elevated NT-proBNP levels (2888 pg/mL in VICTORIA, compared with 1608 and 1437 pg/mL in the PARADIGM-HF and DAPA-HF trials, respectively) and the presence of a higher number of patients in NYHA class III-IV (41% of patients in the VICTORIA trial compared with the 25% and 32% of the patients in the PARADIGM-HF and DAPA-HF trials, respectively) [14,22,25,26]. Future studies should be conducted with a better assessment of patient clinical severity, focusing on clinically stable subjects with advanced HF. This could help identify the best timing and patient’s conditions to start this new drug, enhancing the current guideline algorithm.

Moreover, since the mechanism of action of vericiguat is addressed on endothelial dysfunction restoration, some HFpEF subtypes could benefit from vericiguat administration, particularly those patients with increased vascular stiffness and ventricular–arterial uncoupling evidence. However, in patients with HFpEF, outcomes with vericiguat are still limited. The precise mechanism of action of vericiguat should be tested in detail in further studies by direct cGMP level measurements to confirm the action of the drug and to demonstrate the efficacy of its use in this setting of patients.

## 6. Conclusions

Despite the improvements achieved in recent years with the current treatments, patients with HF are still at high risk of HFH and death during long follow-up periods. The latest ESC/AHA HF guidelines [2,3] recommend vericiguat as a second-line treatment in patients who remain symptomatic despite first-line treatment, manly after a recent decompensation. Based on population characteristics, we believe that this drug could have beneficial effects in chronic stable HF with reduced or mildly reduced ejection fraction, together with the traditional quadruple therapy based on ARNI/ACEi, beta blockers, MRA, SGLT2i and diuretics in those with congestion evidence. The role of vericiguat in more advanced HF stage and in patients with history of recurrent HFH has been previously established. If the drug’s safety can be confirmed in a larger population, its extensive application may become a future challenge to reduce adverse events in multiple HF settings.

## Figures and Tables

**Figure 2 biomedicines-10-02471-f002:**
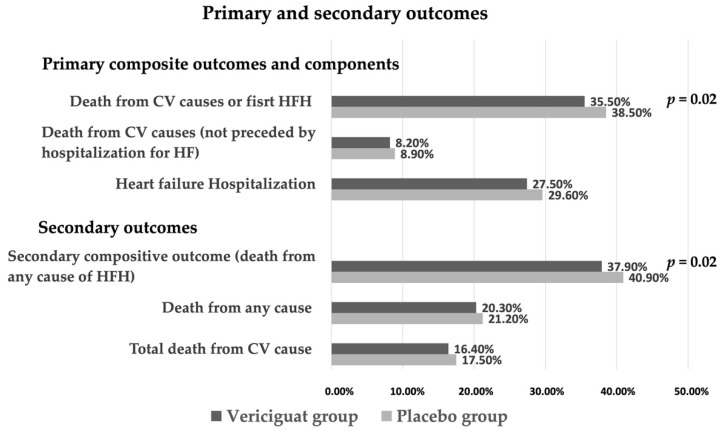
Incidence of primary and secondary outcomes in VICTORIA trial. CV = cardiovascular; HF = heart failure; HFH = heart failure hospitalization.

**Table 1 biomedicines-10-02471-t001:** SOCRATES-REDUCED and VICTORIA trials’ inclusion and exclusion criteria.

	Inclusion Criteria	Main Exclusion Criteria
SOCRATES-REDUCED [18]	Chronic HF (NYHA functional classes II-IV with standard HF therapy ≥ 30 days before HFH), Recent episode of worsening chronic HF defined by: HFH or outpatient administration of intravenous (IV) diuretics within 4 weeks of randomization, signs and symptoms of congestion and elevated natriuretic peptide levels (BNP ≥ 300 pg/mL or NT-proBNP ≥ 1000 pg/mL if in sinus rhythm; BNP ≥ 500 pg/mL or NT-proBNP ≥ 1600 pg/mL if in atrial fibrillation), LVEF < 45% and LA enlargement at randomization.	Clinically unstable (systolic blood pressure < 110 mmHg or >160 mmHg and heart rate < 50/min or >100/min) or receiving IV inotropes at any time after HFH, Cardiac active comorbidity (acute coronary syndrome < 60 days, valvular heart disease, infiltrative myocardial disease, hypertrophic cardiomyopathy with outflow tract obstruction, etc.) Non- cardiac active comorbidity (eGFR < 30 mL/min/1.73 m^2^, hepatic insufficiency as Child-Pugh B or C, obesity with BMI > 40 kg/m^2^, etc.) Concomitant treatment with PDE5 inibitor or sGC stimulator.
VICTORIA [19]	LVEF < 45% assessed within 12 months prior to randomization, BNP ≥ 300 pg/mL or NT-proBNP ≥ 1000 pg/mL if in sinus rhythm (BNP ≥ 500 pg/mL or NT-proBNP ≥ 1600 pg/mL if in atrial fibrillation) within 30 days prior to randomization, Prior HFH within 6 months or outpatient IV diuretic therapy for HF within 3 months prior to randomization.	Clinically unstable, Systolic blood pressure < 100 mm Hg, Clinically active cardiac comorbidity (prior cardiac valve intervention <3 months or coronary revascularization < 60 days), Receiving IV inotropes, an implantable left ventricular assist device or awaiting heart transplantation, Concurrent or anticipated use of long-acting nitrates of sGC stimulator, PDE5 inhibitors, Unable to provide informed consent, Females of reproductive age not using an acceptable form of contraception.

BNP = brain natriuretic peptide; HF = heart failure; HFH = heart failure hospitalization; IV = intravenous; LVEF = left ventricular ejection fraction; NT-proBNP = N-terminal pro-B-type natriuretic peptide; PDE5 = phosphodiesterase type 5; sGC = soluble guanylate cyclase; VICTORIA = Vericiguat Global Study in Subjects With Heart Failure With Reduced Ejection Fraction. Adapted with permission from Ref. [22]. Copyright 2018, Armstrong, P.W. et al.

**Table 2 biomedicines-10-02471-t002:** Summary of the main results data from the SOCRATES-REDUCED and VICTORIA trials.

Trial	Inclusion Criteria	Patients	Treatment	Results
SOCRATES-REDUCED [18]	HF (LVEF < 45%), <4 weeks from HF decompensation	456 (351 patients completed treatment)	Vericiguat (1.25 mg, 2.5 mg, 5 mg, 10 mg daily) vs. placebo	Primary endpoint: no significant difference in NT-proBNP levels from baseline to week 12 (*p* = 0.15).
VICTORIA [19]	HF (LVEF < 45%), NYHA II-IV, BNP ≥ 300 ng/L or NT-proBNP ≥ 1000 ng/L (BNP ≥ 500 ng/L or NT-proBNP ≥ 1600 ng/L if AF), HFH < 6 months or worsening HF requiring iv diuretics < 3 months	5050 (1175 discontinued trial regime but included in the analysis)	Vericiguat (1.25 mg, 2.5 mg, 5 mg, 10 mg daily; 10-mg target dose in 89.2%) vs. placebo	Primary endpoint (CV death or first HFH): HR 0.90 (CI 0.82–0.98). Total HFH: HR 0.91 (CI 0.84–0.99) Death from any cause: HR 0.95 (CI 0.84–1.07) Death from any cause or first HFH (composite secondary outcome): HR 0.90 (CI 0.83–0.98)

AF = atrial fibrillation; BNP = B-type natriuretic peptide; CI = confidence interval; CV = cardiovascular; HF = heart failure; HFH = heart failure hospitalization; HR = Hazard Ratio; LVEF = left ventricular ejection fraction; NT-proBNP = N-terminal pro-B-type natriuretic peptide; NYHA = New York Heart Association. Adapted with permission from Ref. [20]. Copyright 2021, Lombardi CM, et al.

**Table 3 biomedicines-10-02471-t003:** Subgroup analysis of the VICTORIA trial.

Characteristic		Number of Patients (Vericiguat/Placebo)	Hazard Ratio 0.5–1.0 (Better Vericiguat) 1.0–1.5 (Better Placebo)	95% Confidence Interval
Age				
	<65 yr	290/348	0.81	(0.70–0.95)
	≥65 yr	607/624	0.94	(0.84–1.06)
	<75 yr	579/669	0.84	(0.74–0.94)
	≥75 yr	318/303	1.04	(0.88–1.21)
Index event				
	IV diuretics < 3 months ago	96/120	0.78	(0.60–1.02)
	HFH < 3 months ago	660/701	0.93	(0.84–1.04)
	HFH 3–6 months ago	141/151	0.85	(0.67–1.07)
Baseline NYHA class				
	I–II	445/484	0.91	(0.80–1.04)
	III–IV	451/487	0.87	(0.77–0.99)
Use of sacubitril/valsartan				
	Yes	134/153	0.88	(0.70–1.11)
	NO	760/818	0.90	(0.81–0.99)
Baseline eGFR				
	≤30 mL/min/1.73 m^2^	143/128	1.06	(0.83–1.34)
	>30 to ≤60 mL/min/1.73 m^2^	392/455	0.84	(0.73–0.96)
	>60 mL/min/1.73 m^2^	346/372	0.92	(0.80–1.07)
Baseline NT-proBNP				
	Quartile 1 (≤1556.0 pg/mL)	128/161	0.78	(0.62–0.99)
	Quartile 2 (>1556.0 to ≤2816.0 pg/mL)	165/201	0.73	(0.60–0.90)
	Quartile 3 (>2816.0 to ≤5314.0 pg/mL)	213/257	0.82	(0.69–0.99)
	Quartile 4 (>5314.0 pg/mL)	355/302	1.16	(0.99–1.35)
Baseline LVEF				
	<35%	637/704	0.88	(0.79–0.97)
	≥35%	255/265	0.96	(0.81–1.14)
	<40%	773/851	0.88	(0.80–0.97)
	≥40%	119-117	1.05	(0.81–1.36)

eGFR= glomerular filtration rate; HFH = heart failure hospitalization; IV = intravenous; LVEF = left ventricular ejection fraction; NT-proBNP = N-terminal pro-B-type natriuretic peptide; NYHA = New York Heart Association. Adapted with permission from Ref. [19]. Copyright 2020, Armstrong, P.W., et al.

**Table 4 biomedicines-10-02471-t004:** Summary of the main results data from the SOCRATES-PRESERVED and VITALITY trials.

Trial	Inclusion Criteria	Patients	Treatment	Results
SOCRATES-PRESERVED [30]	HFpEF (LVEF ≥ 45%), <4 weeks from HF decompensation NYHA II-IV, BNP ≥ 100 ng/L or NT-proBNP ≥ 300 ng/L (BNP ≥ 200 ng/L or NT-proBNP ≥ 600 ng/L if AF), Left atrial enlargement.	477 (325 patients completed treatment)	Vericiguat fixed-dose treatment arms (1.25 mg or 2.5 mg) and vericiguat up-titrated treatment arms (2.5–5 mg, 2.5–10 mg daily) vs. placebo	Primary endpoint: no significant difference change in NT-proBNP levels (*p* = 0.20) and change in LAV (*p* = 0.37) from baseline vs. placebo to 12 weeks. Exploratory endpoint: improve KCCQ score change from baseline (mean difference: 19.8 points) vs. placebo (mean difference: 9.2 points) to 12 weeks.
VITALITY [31]	HFpEF (LVEF ≥ 45%), NYHA II-III, HFH or HF requiring iv diuretics < 6 months, BNP ≥ 100 ng/L or NT-proBNP ≥ 300 ng/L (BNP ≥ 200 ng/L or NT-proBNP ≥ 600 ng/L if AF), Left ventricular hypertrophy or left atrial enlargement.	789 (761 included in primary analysis)	Vericiguat (up-titrated to 15 mg or 10 mg) vs. placebo	Primary endpoint: no significant change in KCCQ to 24 weeks. Secondary outcome: no difference in 6MWT to 24 weeks.

6MWT = six-minute walk test; AF = atrial fibrillation; BNP = B-type natriuretic peptide; HFH = heart failure hospitalization; HFpEF = heart failure with preserved ejection fraction; KCCQ = Kansas City Cardiomyopathy Questionnaire; LAV = left atrial volume; NT-proBNP = N-terminal pro-B-type natriuretic peptide; NYHA = New York Heart Association.

## Data Availability

Not applicable.

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
