# Peer review of "Vericiguat in Heart Failure: Characteristics, Scientific Evidence and Potential Clinical Applications"

_biomedicines, 2022, doi:10.3390/biomedicines10102471_

Round 1
Reviewer 1 Report
The review comprehensively summarized the drug characters and the results of RCT of vericiguat for heart failure. It is timely and useful information. The review is well written. RCT for both HFREF and HFPEF were included.
The reviewer recommended the suggestions to add two perspective for the review, which could be improved and increase interests for readers.
a) The opinion for the clinical usage for vericiguat. From RCT, vericiguat could be useful for severe symptomatic HFREF patients for the improvement of symptoms. For the future, can which kind of patients be the target for the usage of this drug? Because recently many of drugs for HFREF were developed (SGLT-2 ARNI b-blockers MRA), how is the timing of using vericigat. Currently it may not be effective for HFREF, however there may be possibility.
b) NO activated both cGMP-dependent pathway and independent pathway (targeting such as SERCA2, potassium channels and Na K ATPase). How is the other molecular target of NO for HF and the drug development for these targets. ARNI activated particular GC. The difference of pGC and sGC might be discuss in the review.
Reviewer 2 Report
The idea behind the mansucript is ok, the scientific information is good, but the text has to be reorganized to be more readable, the sentences in many places are too redundant.
Reviewer 3 Report
The manuscript "Vericiguat in Heart Failure: characteristics, scientific evidence, and potential clinical applications " overviews the potential application and pharmacological characteristics of vericiguat (a drug that acts in the cyclic guanosine monophosphate (cGMP) pathway through direct and indirect stimulation of soluble guanylate cyclase) in HF with a specific focus on the clinical trials investigating clinical outcomes and the mechanisms of vericiguat in HFrEF and HFpEF.
This review is informative and would be interesting for potential readers after extensive revision to improve logical flow, provide missing references, as well as English editing to improve readability. Several examples of necessary improvements (but not limited to) are summarized below.
Please use HF, HFrEF, etc., consistently as abbreviated after the first introduction.
Vericiguat must be consistently used capitalized or non-capitalized.
Abstract:
1st sentence should be corrected as follows: “Although recent advances in the management of heart failure (HF), the risk of death and hospitalization remains high during long follow-ups.”
2nd sentence is incorrect since the same statement applies not only to HFrEF but also to HFpEF.
Consider something like - “HF (regardless of ejection fraction) is characterized by endothelial dysfunction, inflammation, and increased oxidative stress associated with a reduction in the activity of the nitric oxide (NO)-soluble guanylate cyclase (sGC)-cyclic guanosine monophosphate (cGMP) signaling pathway, leading to damaging effects on the myocardial, vascular and renal levels. Vericiguat repristinates the relative deficiency of this signaling pathway through stimulation and activation of sGC, increasing cGMP levels and related signaling, and may attenuate HF-related oxidative stress and endothelial dysfunction. ” And then describe outcomes of clinical trials in patients with HFrEF or HFpEF.
Line 16: word “trial” is missed.
The last sentence should be, “In the current paper, we reviewed the potential application and pharmacological characteristics of vericiguat in HFrEF and HFpEF.”
Introduction:
Line 29: An estimated 64.3 million people have HF worldwide.
- Groenewegen, A.; Rutten, F.H.; Mosterd, A.; Hoes, A.W. Epidemiology of Heart Failure. Eur J Heart Fail 2020, 22, 1342-56.
2. Mechanism of drug action:
1st paragraph (lines 53-56): references must be provided to support each statement.
Line 53: “Stimulates” instead of “acts.”
Lines 54-56: This statement does not align with the Scheme presented in Fig.1.
2nd paragraph should be focused on the NO-sGC-cGMP pathway. Therefore, I would suggest removing the 1st sentence as confusing and misleading. 2nd sentence (lines 62-63) should be truncated as “The NO-sGC-cGMP pathway begins with NO production by vascular endothelial cells.”
Line 63-65: re-write as “NO is synthesized from L-arginine by three nitric oxide synthase, among which endothelial nitric oxide synthase (eNOS) plays a major role.”
Must cite original publications instead of ref. 6.
The 3rd (lines 73-81) and 4th (lines 84-90) are redundant and must be combined in one logical setting.
Lines 91-92: Completely unclear what the authors wanted to say.
Figure 1:
- - Figure legend must provide a short description of the Scheme and an explanation of the abbreviated symbols.
- - “Relative cGMP deficiency” instead of “cGMP deficiency”.
- - Increase size of “GTP” and “cGMP”, which are currently unreadable.
- - I would strongly recommend removing the images of cells since they do not add meaning value but are somewhat misleading.
3. Pharmacology proprieties of Vericiguat
The sub-title must be corrected as “Pharmacological proprieties of Vericiguat.”
Line 95: reference is required.
Please provide the drug doses supported by references.
Paragraphs 2, 3, and 4 need to be re-written and restructured in one logical and not redundant text.
Line 111: “are not inhibited?”
4. Main clinical Trials
Better it should be “Main HF Clinical Trials”.
Line 167: “However” instead of “Furthermore, we can add that.”
Table 1: Please include the inclusion and exclusion criteria of the phase II study SOCRATES-REDUCED to this table.
Line 282: “examines” instead of “examinate”.
5. Vericiguat in HFpEF: should be sub-section 4.2 instead of 5.
Lines 331-333: Seems the clinical trials in HFrEF have same limitations (lack of cGMP measurements). This point must be clearly addressed in 4.1 subsection.
Lack of cGMP measurements should also be addressed in Conclusions.
6. Conclusions: should be 5 instead of 6.
Conclusions may be re-written for clarity and focus.
Round 2
Reviewer 2 Report
The text was significantly improved structurally and scientifically, and is of interest for those involved in HF management.
Still the English of the article has to be improved. Please use professional copy editing or adopt a native English speaking co-author/coworker.
Reviewer 3 Report
Line 17: Please change “increasing cGMP levels....” to “aiming to increase cGMP levels..., since clinical trials did not prove this fact.
Figure 2: all texts are unreadable; please increase font size significantly for readability.
